# Oxytocin Differentiated Effects According to the Administration Route in a Prenatal Valproic Acid-Induced Rat Model of Autism

**DOI:** 10.3390/medicina56060267

**Published:** 2020-05-29

**Authors:** Radu Lefter, Alin Ciobica, Iulia Antioch, Daniela Carmen Ababei, Luminita Hritcu, Alina-Costina Luca

**Affiliations:** 1Center of Biomedical Research, Romanian Academy, B dul Carol I, No 8, 700505 Iasi, Romania; radu_lefter@yahoo.com; 2Department of Research, Faculty of Biology, Alexandru Ioan Cuza University, B dul Carol I, No 11, 700506 Iasi, Romania; iulia.antioch@gmail.com; 3“Grigore T.Popa” University of Medicine and Pharmacy, 16, Universitatii Street, 700115 Iasi, Romania; dana.ababei@gmail.com (D.C.A.); acluca@yahoo.com (A.-C.L.); 4Faculty of Veterinary Medicine, University of Agricultural Sciencies and Veterinary Medicine “Ion Ionescu de la Brad” of Iasi, 3rd Mihail Sadoveanu Alley, 700490 Iasi, Romania

**Keywords:** autism, oxytocin, intranasal, intraperitoneal, oxidative stress, gastrointestinal disturbances

## Abstract

*Background and objectives*: The hormone oxytocin (OXT) has already been reported in both human and animal studies for its promising therapeutic potential in autism spectrum disorder (ASD), but the comparative effectiveness of various administration routes, whether central or peripheral has been insufficiently studied. In the present study, we examined the effects of intranasal (IN) vs. intraperitoneal (IP) oxytocin in a valproic-acid (VPA) autistic rat model, focusing on cognitive and mood behavioral disturbances, gastrointestinal transit and central oxidative stress status. *Materials and Methods*: VPA prenatally-exposed rats (500 mg/kg; age 90 days) in small groups of 5 (*n* = 20 total) were given OXT by IP injection (10 mg/kg) for 8 days consecutively or by an adapted IN pipetting protocol (12 IU/kg, 20 μL/day) for 4 consecutive days. Behavioral tests were performed during the last three days of OXT treatment, and OXT was administrated 20 minutes before each behavioral testing for each rat. Biochemical determination of oxidative stress markers in the temporal area included superoxide dismutase (SOD), glutathione peroxidase (GPx) and malondialdehyde (MDA). A brief quantitative assessment of fecal discharge over a period of 24 hours was performed at the end of the OXT treatment to determine differences in intestinal transit. *Results*: OXT improved behavioral and oxidative stress status in both routes of administration, but IN treatment had significantly better outcome in improving short-term memory, alleviating depressive manifestations and mitigating lipid peroxidation in the temporal lobes. Significant correlations were also found between behavioral parameters and oxidative stress status in rats after OXT administration. The quantitative evaluation of the gastrointestinal (GI) transit indicated lower fecal pellet counts in the VPA group and homogenous average values for the control and both OXT treated groups. *Conclusions*: The data from the present study suggest OXT IN administration to be more efficient than IP injections in alleviating autistic cognitive and mood dysfunctions in a VPA-induced rat model. OXT effects on the cognitive and mood behavior of autistic rats may be associated with its effects on oxidative stress. Additionally, present results provide preliminary evidence that OXT may have a balancing effect on gastrointestinal motility.

## 1. Introduction

Autism is a lifelong developmental neurologic disorder characterized by deficits in social communication and interaction, restricted, repetitive patterns of behavior, interests, or activities, originating in the early developmental period. Cognitive deficits and atypicalities appear to be a characteristic albeit not a universal feature [1] and are situated in the social and communicative range, including verbal and nonverbal expression [2,3], emotional processing and social reciprocity [4,5]. Deficits in memory function, an important component of cognitive domain, have been inconsistently reported, but, while basic processes seem to remain intact—or even potentiated, such as the mechanical memories related to numbers, names etc., defects of working-memory manifest in more complex tasks that require multiple information to be integrated and recalled [6].

Different comorbidities occur often along with the core symptoms [7], such as gastrointestinal disorders, auditory disorders and infections, or psychiatric disorders that may severely affect daily normal functioning [8]. Among the latter, symptoms of anxiety or depressive disorders are consistently described in three quarters of affected children [9]. The correlations between GI symptoms and ASD behavioral phenotype carefully revised in a very systematical meta-analysis seem to relate to alterations of gut microbiota composition [10]. Dinan and Cryan suggest microbiome to be implicated in ASD etiology [11] and cite a series of studies on germ-free animals that show the impact of microbiome during neurogenesis period on amygdala [11,12,13] and hippocampus [11,14] critical brain areas for anxiety and fear-related behavior and memory respectively [11].

Valproic acid (VPA) and its salt derivates, commonly used as anticonvulsants, antimigrainic and mood-stabilizing drugs [15], are also known for their teratogenic effects when used during pregnancy, and increased risk for autistic symptoms in the developing fetus [16]. The functional and structural abnormalities discovered in VPA prenatally exposed rodents at the brain level are characterized by abnormal cerebral early overgrowth and increased local synaptic plasticity and connectivity, cerebellar abnormalities and lowered number of Purkinje cells which are similar to those characteristic to the autistic human brain [15,17,18]. Currently, the prenatally VPA exposed animal model is the most used reliable model to study autistic pathology, as one single administration of VPA to pregnant rat or mouse females on the 12.5th embryonic day is sufficient to induce in the offspring decreased sociability, increased repetitive behaviors and anxiety and other specific autistic-like traits [19], along with the above mentioned neuro-anatomical modifications. The 12.5th day of the first trimester of gestation corresponds to neural tube enclosure and brain stem nuclei genesis and thus is a particular vulnerable period in embryogenesis leading to defects of neural tube and congenital malformations [20].

Increased oxidative stress is reported by numerous studies finding high concentration of lipid peroxidation products, decreased activity of antioxidant enzymes (superoxide dismutase, glutathione peroxidase, catalase) and significant lower Total Antioxidant Status in plasma of autistic patients [21,22,23] and in key regions of brain of autistic individuals, such as temporal cortex [24], cerebellum [25] or hippocampus [26,27].

Actually, more than just an endogenous subsequent imbalance, oxidative stress could be one of the principal factors in the pathogenesis of the disorder together with immune dysregulation and inflammation and mitochondrial dysfunction [28,29]. Several studies using Western blotting and immunohistochemistry to investigate the mechanisms by which VPA exerts its teratogen effects in cell cultures or whole-embryo culture models indicate increased fetal oxidative stress marked by significant production of reactive oxygen species (ROS) in the head region correlated with increased apoptotic markers [30].

The hormone oxytocin (OXT) plays an important role modulating social behavior, with projections and receptors densely distributed in specific brain areas responsible of social perception, cognition and social anxiety, including amygdala, hippocampus and VTA [31,32]. Abnormalities in OXT signaling pathways are relevant to autistic etiology. Polymorphisms of genes encoding the proteins and related regulatory factors such as the OXT-NPI (encoding the OXT prepropeptide), OXTR or CD38 genes (involved in OXT release in the brain), are associated with social deficits and repetitive behavior and have been reported in autistic populations [33,34]. Similarly, findings in animal models relate impairments of the OXT system to ASD symptoms such as social memory [35] restricted interests to novelty [36] or social communication and interaction [37,38].

OXT has potent anti-inflammatory and antioxidant properties, related to its structure containing tyrosine and tryptophan residues, inhibitors of lipid peroxidation and oxidative apoptosis [39,40]. This may explain the homeostasis-maintaining action of OXT observed in diet-induced obesity rat model [41], atopic dermatitis [42], atherosclerosis [43].

Very few studies have focused on (the link between) compromised antioxidant system in autism and OXT, and mechanisms underlying oxytocin deficiency are not elucidated, whether originating in the brain or involving immune signaling dysregulation along the gut/brain system presumptively linked with perinatal hypoxic stress [44]. However, recent reports of postnatal OXT treatment on VPA-induced animal models of autism are very promising demonstrating long-term effects ameliorating the social impairments and elevated oxidative stress and inflammation [38,45].

Whereas IP administration of OXT has been used with positive results in rodents, for treatment of abnormal social behavior in *Cntnap2* knocked out mice [46] or to increase OXT immunoreactivity and social contact in prairie voles [47,48], the non-invasive intranasal (IN) administration may be a more efficient way in approaching central nervous system (CNS) diseases, as this route was recently reported to allow the neuropeptide to bypass the blood-brain barrier and permeate into specific areas of the brain [49]. The direct access to the CSF has the advantage of reaching effective concentrations in short time, installment of behavioral effects that last several hours, unlike the short peripheral half-life, and avoidance of systemic side effects [31,50]. Therefore, IN administration, which has already been reported as a very effective route for therapeutic outcome [51] could be more beneficial for brain processes than peripheral IP administration, an aspect which needs further investigation.

In the present study, we used the VPA prenatal exposure rat model of autism to investigate the effects of OXT on its behavioral phenotype with focus on autistic comorbidities, including anxiety, depression and cognitive impairment, and the oxidative stress level in the brain. Secondly, we used two different routes of OXT administration, IP and an original IN per-nostril adaptation. Few studies have focused on the implications of VPA exposure on the functionality of GI tract and OXT effects on intestinal peristaltism and gastric emptying, yet recent results on VPA animal models suggest a more complex picture with digestive disturbances and microbiome modifications accompanying the cognitive and social deficits [52,53]. Thus, as a secondary objective of the study, we were interested if differences in intestinal transit suggestive for GI disturbances would occur between groups.

## 2. Materials and Methods

### 2.1. VPA Exposure and OXT Treatment

White Wistar pregnant female rats (*n* = 9) weighing around 200 g were used in this experiment. They were housed in room temperature of (22 ± 2 °C) at 50% ± 10% relative humidity, 12 h light-dark cycle, with free access to food pellets and water. On the 12.5th gestational day the pregnant rats were randomly divided into two groups: VPA group (*n* = 6) and control group (*n* = 3). VPA group was given one single intraperitoneal (IP) injection of sodium salt of valproic acid (Na VPA, Sigma-Aldrich, Saint-Louis, MO, USA) in 0.9% saline (100 mg/mL) at a dose of 500 mg/kg bodyweight. Control group received a single IP injection of water/saline 0.9%). The dams were subsequently housed individually and allowed to raise their litters. The offspring were weaned on postnatal day (PN) 21, separated according to gender and only male offspring further selected for the study and housed in groups of two–three animals per cage in a room under controlled humidity and temperature, with food and water ad libitum.

Twenty pups were selected and four groups (each with *n* = 5) were created for the postnatal experiments: the control group (I) that was exposed only to saline, the VPA group (II), and two other in utero VPA exposed groups that underwent a sub-chronic OXT treatment, the IP OXT-VPA group (III), that received IP OXT injection, and the IN OXT-VPA group (IV) receiving intranasal per-nostril OXT. Behavioral effects of IP oxytocin in rats have been previously studied by our group [45]. The timeline for protocol and behavioral testing is detailed in Figure 1.

OXT treatment began at PND90 when rats reached an average weight of 150 g. OXT (Sigma-Aldrich Co., LLC., Darmstadt, Germany) was intraperitoneally injected in the IP OXT-VPA group in a dose of 10 mg/kg/bodyweight for 8 days consecutively. This dosage was established based on previous published work of our group [54,55] and others [56]. Behavioral tests were initiated on the 6th day of OXT treatment and lasted three days during which the OXT administration continued. For each rat OXT was IP injected 15 min before the behavioral test. Controls received only intraperitoneally 0.9% saline given at the same volume of injection as the oxytocin dose.

In the experimental procedure for IN administration, animals were progressively accustomed over a shortened period of one week to a simulated administration with a P10 pipette tip on the rhinarium, while being immobilized, in order to avoid and minimise excessive struggling and head shaking. The advantage of this approach is that it allows efficient IN administration of the oxytocin and does not require prior sedation. For the IN administration, we used 12 I.U./kg bodyweight, following a recent study of Zoratto et al. [57]. Very recently, Pagani et al., 2019, showed that 0.3 I.U. of IN OXT under both acute and chronic administration reaches brain and elicits substantial effects on brain connectivity [58]. Rats were administered 20 μL of OXT (at a concentration of 15 µg/mL) once daily, for 4 consecutive days, with a P10 pipette. Alternating drops of OXT were applied on each nostril at 15 s intervals, to ensure OXT absorption. Between each round of two alternative drops pipetting, rats were given a short break for recovery. Behavioral tests were performed during the last three days of OXT treatment. As previously, IN OXT was administrated 15 min before each behavioral testing for each rat.

All the experiments were performed in the light phase between 09:00 and 17:00. Animals treatment and maintenance were carried out in accordance with the current national and European Regulations regarding the scientific research using animals and in accordance with NIH-Care and Use of Laboratory Animals Manual (8th Edition) and in accordance with the guidelines of animal bioethics from the Act on Animal Experimentation and Animal Health and Welfare Act from Romania and all procedures were in compliance with the European Communities Council Directive of 24 November 1986 (86/609/EEC). This study was approved by the local Faculty Ethics Committee (USAMV no. 518/05.06.2019) and also efforts were made to minimize animal suffering and to reduce the number of animals used.

### 2.2. Behavioral Testing

The behavioral tests used in this experiment were intended for assessing memory deficit, anxiety and depression-like behaviors, suggestive for comorbid impairments often signaled in ASD: Y-maze, elevated-plus maze and forced swimming test.

### 2.3. Y-Maze test

The Y-maze task is a reliable test to determine the memory performance changes based on measuring the spontaneous alternation behavior as a marker for the short-term memory. Short-term memory was assessed by spontaneous alternation behavior in the Y-maze task. The methodology has been previously described in a recent study of our group [59].

### 2.4. The Elevated Plus Maze Test (EPM)

The elevated plus-maze, a standard paradigm to assess anxiety-related behavior in rodents, consists of four arms, 49 cm long and 10 cm wide, elevated 50 cm above the ground, with two arms enclosed by walls of 30 cm high and the other two exposed. After accommodation to the test room for half an hour, each rat was placed at the junction of the four arms, facing an open arm and allowed to freely explore the maze for 5 min. The entries and time duration in each arm were recorded as indicators for anxiety. Other specific ethological parameters such as head dipping in open-arms, protected stretch-attend postures in the closed arms and grooming bouts were also measured.

### 2.5. Forced Swim Test

For this test used for evaluating depressive-like states, rats were individually placed into a cylindrical recipient (diameter 30 cm, height 59 cm) containing 25 cm of water at 24 ± 1 °C in order to assess escape behavior. We have used the shortened protocol version of the test, consisting of 6 min swimming sessions for each rat, first two minutes for acclimatization and the last 4 min for measuring. Two indicators were considered: floating time (including minimal movements to keep the head above the water), as an indicator of depressive state.

### 2.6. Intestinal Transit Assessement

A quantitative assessment of fecal discharge over a period of 24 h was performed at the end of the OXT treatment, prior to final tissue prelevation. Each group of rats was placed into clean normal cages after OXT treatment and last behavioral test and kept for 24 h with water and food ad libitum, in order to avoid significant coprophagy. After 24 h, the faecal boli from each cage were collected and counted.

### 2.7. Tissue Collection

At the end of the behavioral testing, rats were anesthetized and sacrificed using an anesthetic mix of Ketamine/Xylazine and whole brains were removed. The temporal lobes were collected for oxidative stress assays and homogenized with a Potter Homogenizer coupled with Cole-Parmer Servodyne Mixer in bidistilled water (1 g tissue/10 mL bi-distilled water). Following centrifugation (15 min at 3000 rpm), the supernatant was separated and pipetted into tubes.

### 2.8. Biochemical Analysis

Biochemical determination of oxidative stress level in the brain was carried and included quantification of total content of malondialdehyde (MDA) and glutathione peroxidase (GPx) and superoxide dismutase (SOD) specific activities.

Superoxide dismutase (SOD) activity was measured by the percentage reaction inhibition rate of enzyme with WST-1 substrate (water-soluble tetrazolium dye) and xanthine oxidase using a SOD Assay Kit (Sigma-Aldrich, Saint Louis, MO, USA) as previously described [60].

Glutathione peroxidase (GPx) activity was measured using the GPx cellular activity assay kit CGP-1 (Sigma-Aldrich, Saint Louis, MO, USA). The method was previously described [61] and uses an indirect determination based on the oxidation of glutathione to oxidized glutathione coupled with the inverse reaction in the presence of GPx, glutathione reductase, and nicotinamide adenine dinucleotide phosphate (NADPH) as an enzymatic cofactor. The method is based on the NAPDH concentration decrease measurement in the reaction media, correspondent to the GPx activity during which NADPH is oxidized to NADP+.

MDA concentrations were determined by thiobarbituric acid reactive substances assay, as previously described [60]. Temporal lobe homogenate (supernatant, 200 μL) was added and briefly mixed with 1 mL of trichloroacetic acid at 50%, 0.9 mL of Tris-HCl (pH 7.4) and 1 mL of thiobarbituric acid 0.73%. After vortex mixing, samples were maintained at 100 °C for 20 min. Samples were centrifuged at 3000 rpm for 10 min, and supernatant was read at 532 nm.

### 2.9. Statistical Analysis

Statistical analysis tests using Minitab 17 (Minitab Inc., State College, PA, USA, 2013) application to compare the differences between groups included one-way analysis of variance (ANOVA). Post hoc analysis were performed using Tukey’s honestly significant difference test in order to compare groups. All results are expressed as the means ± SEM. Results were considered of statistical significance at a value of *p* < 0.05. Pearson’s correlation coefficient and regression analysis were used to evaluate the connection between oxidative stress markers and behavioral parameters.

## 3. Results

### 3.1. Spontaneous Alternation Percentage in the Y-Maze Task

The percentage of spontaneous alternation (spatial memory) of rats in the Y maze is represented by Figure 2A. A significant difference between the percentage of spontaneous alternation of the different groups was indicated by one way ANOVA (F (3.20) = 6.85; *p* = 0.0060). Post hoc comparisons showed that VPA induced a significant decrease in spontaneous alternation in the prenatally exposed rats, when compared to control group (*p* < 0.01). This significant deficit in spontaneous alternation behavior persisted in the VPA-exposed rats even after IP OXT treatment when compared to control (*p* < 0.05), but the statistical difference was less pronounced than in the untreated VPA-exposed only group. On the other hand, for the animals intranasally treated with OXT, we noticed a significant increase of the spontaneous alternation parameter as compared to the VPA-exposed animals (*p* < 0.01), and similar performance as the control group. Also, there was no significant difference in the locomotor activity (as judged by the number of arms entries) in any of the four groups.

### 3.2. Effects of Oxytocin on Anxiety-Like Manifestations in the Elevated Plus Maze Task

#### 3.2.1. Time Spent in the Opened Arms of the EPM

Regarding time spent in the opened arms by rats, we observed a significant group difference (F (3.20) = 3.785; *p* = 0.040) (Figure 2B). Post hoc comparisons showed that rats of the VPA-exposed group spent significantly decreased time in the open arms when compared to that of control group (*p* < 0.05). OXT treatment both under IN and IP regimen increased time in the opened arms in the previously VPA-exposed animals to a level where differences vs. control were non-significant. However, differences vs. VPA group remained also non-significant (Figure 2B).

#### 3.2.2. Number of Entrances in the Opened and Closed Arms of the EPM

We also observed a significant overall effect of the treatment on the number of entrances in the opened (F (3.20) = 3.138; *p* = 0.0481) and closed arms of the EPM (F (3.20) = 7.210; *p* = 0.0061). Additionally, post hoc analysis revealed that prenatal exposure to VPA reduced the number of entries in the opened arms and, surprisingly, in the closed arms also, in a significant way when compared to the control group (*p* < 0.05). Treatment with OXT both by IN and IP administration increased the number of entrances in the opened arms and closed arms in the VPA in utero-exposed animals to a level where differences vs. control were non-significant. The treatment with IN OXT significantly increased (*p* < 0.01) the number of entries in the closed arm when compared to the VPA group (Figure 2C).

#### 3.2.3. Number of Head Dips in the Opened Arms of the EPM

We observed significant modifications concerning head dipping behavior in the EPM in the experimental groups (F (3.20) = 3.9336; *p* = 0.0393). Head dipping at the extremities of the open arms is another important ethological parameter in the EPM with relevance to exploratory behavior and anxiolytic activity. The significant decrease of this parameter in the VPA-prenatally exposed group (*p* < 0.05 vs. control) can be linked to the similarly decreased time spent in the open arms (Figure 2B). The OXT administration reversed the apparent anxyogenic effects of VPA increasing the number of head dips in the open arms up to a significant extent for the IP OXT group when compared to the VPA group (*p* < 0.05) (Figure 2C).

For other ethological parameters measured in the EPM, such as stretching postures and number of grooming bouts, no significant differences were recorded between groups.

### 3.3. Effect of Oxytocin on Depressive-Like Manifestations in the Forced-Swimming Task

The depressive-like response was assessed as the time spent floating in immobility or with minimal movement. Statistical analysis of the immobility time showed a significant difference between all groups (F (3.20) = 22.27; *p* < 0.0001) (Figure 2E). Post hoc comparisons revealed prenatal VPA-exposure resulted in a very significant increase of immobility time in the VPA group (*p* < 0.001 vs. control). After OXT administration via IP route, depressive manifestations persisted in the VPA-exposed rats as significantly prolonged periods of immobility (*p* < 0.005 vs. control). The OXT IN treatment revealed however antidepressive effects and resulted in significantly decreased floating time (*p* < 0.05 vs. VPA group), although relatively increased when compared to control (*p* < 0.01) (Figure 2E).

### 3.4. Effect of Oxytocin on Oxidative Stress Biomarkers

We found a significant decreased SOD activity in prenatally VPA-exposed group (*p* < 0.001 vs. control) when compared to control group. IP OXT determined an increase of SOD activity when compared to VPA-exposed group (*p* < 0.05 vs. VPA group), but did not attenuated the significant decrease vs. the control group (*p* < 0.01 vs. control). IN OXT resulted in a marked increase of SOD activity when compared to the VPA-only exposed group (*p* < 0.01 vs. VPA group), and also repaired the difference vs. the control group (*p* < 0.05) (Figure 3A).

GPx activity of the temporal lobes was significantly impaired in the rats prenatally exposed to VPA, compared to control rats (*p* < 0.0005). OXT IN administration to VPA exposed rats resulted in a marked increase of GPx activity when compared to the VPA-only exposed group (*p* < 0.05 vs. VPA group), and also attenuated the difference vs. the control group (*p* < 0.05). Interestingly, after OXT IP administration to VPA exposed rats we did not found any clear significant modifications of the GPx activity when compared to the control group—as well as to the VPA-only exposed group (Figure 3B).

When analyzing the levels of lipid peroxidation, we observed a significant increase of MDA levels in the VPA group when compared to the control group (*p* < 0.01). Regarding the effects of OXT, ANOVA revealed a decrease of MDA levels for both IP and IN groups, for which we did not found any significant difference when compared to control group. However, antioxidant action was more visible in the IN OXT group for which MDA levels were significantly decreased when compared to the VPA exposed group (*p* < 0.05) (Figure 3C).

### 3.5. Pearson Correlations

When we performed the Pearson correlations (Table 1) between the behavioral parameters which we determined (spontaneous alternation percentage in Y-maze/mobility in forced swimming task and open arms time, closed arms entries and head dipping in elevated plus maze) and the oxidative stress markers, we obtained significant correlations for the following: spontaneous alternation vs. GPX (*n* = 20, r = 0.660, *p* = 0.002), spontaneous alternation vs. SOD (*n* = 20, r = 0.538, *p* = 0.014), spontaneous alternation vs. MDA (*n* = 20, r = −0.727, *p* < 0.001), for mobility vs. SOD (*n* = 20, r = 0.750, *p* < 0.001), mobility vs. GPX (*n* = 20, r = 0.446, *p* = 0.049) or mobility vs. MDA (*n* = 20, r = −0.449, *p* = 0.047), and also for open arms time vs. SOD (*n* = 20, r = 0.504, *p* = 0.023), or open arms time vs. MDA (*n* = 20, r = −0.387, *p* = 0.092). However, we did not find a significant correlation for open arms time vs. GPX (*n* = 20, r = 0.288, *p* = 0.192), and also for closed arms entries and head dipping vs. oxidative stress markers, which may be the result of exacerbated freezing behavior displayed by some animals during the task. Similar relationship were found when we performed simple regression analysis (Table 2).

### 3.6. Effects of VPA Prenatal Exposure and Oxytocin Treatment on the Intestinal Transit

As mentioned, a quantitative assessment of fecal discharge over a period of 24 h was performed at the end of the OXT treatment, prior to final tissue prelevation. Each group of rats was placed into clean normal cages after OXT treatment and last behavioral test and kept for 24 h with water and food ad libitum, in order to avoid significant coprophagy. After 24 h, the faecal boli from each cage were collected and counted. The fecal pellet counts in the VPA group was much lower than the control group, at an average value of 35.6; an average of 54.4 was obtained for the control group, 47 for the IP OXT group and 45.6 for the IN OXT group. As these were overall values for the entire groups, no significant differences were calculated between groups. However, our data could suggest that whereas VPA exposure has longtime effects on intestinal transit and may cause constipation, OXT treatment seems to increase or balance bowel movements and fecal discharge with no concern to the administration route.

## 4. Discussion

In the present study, we comparatively examined for the first time the effects of central vs. peripheral administrated OXT on cognitive and mood behavioral disturbances in the VPA rat model, and also on oxidative stress status in the temporal brain. Summarizing our results, we observed that (1) sub-chronic OXT treatment generally diminishes the behavioral deficits that occur after VPA prenatal exposure in rats, however (2) IN treatment had significantly better outcome than IP OXT in improving short-term memory reflected by the spontaneous alternation performance in the Y-maze task, (3) or in alleviating depressive manifestations and decreasing floating time in the forced swimming test. However, (4) improvement in the anxiety-like behavior in the elevated plus maze was analogous for both IN and IP paths, when considering the most relevant indicators, the time spent in open arms and number of entrances in the opened and closed arms, whereas exploratory behavior as reflected by number of head dipping was significantly increased only following the IP OXT treatment. (5) Furthermore, we found that whereas VPA causes in prenatally exposed rats increased oxidative stress in the temporal lobe tissue, as reflected by decreased antioxidant enzymatic activity and high lipid peroxidation, OXT treatment showed a positive anti-oxidative effect. Interestingly, IN treatment led to a much clearer mitigation of lipid peroxidation significantly decreasing MDA levels in the temporal lobes, effect less pronounced in the case of the IP treatment, also IN treatment resulted in a more consistent antioxidant GPx enzymatic activity. (6) Finally, after quantitative evaluation of the GI transit in the present model, despite the limitations of the method, we found (7) that VPA exposure caused disturbed intestinal transit reflected by a much lower fecal pellet count over 24 h than the control group. (8) In the OXT treated groups, values of fecal pellet count were sensibly increased suggesting that OXT may have a balancing effect on gastrointestinal motility.

As there is already growing knowledge on evidencing OXT relevance in improving social behaviors [51,62] we chose to focus the behavioral testing on the various comorbidities others than social deficits, as, in our view, the diversity of autistic impairments should be also addressed by a therapeutic agent.

There is a growing view that working memory deficits, that involve cognitive domains involved in social impairments, communication problems, and repetitive activities ([63] as cited by [64]), may be highly relevant in ASD [64]. Although not unanimously signaled in autistic individuals, majority of neuropsychological studies clearly indicate at least some degree of deficit in spatial and spatial-visual working memory tasks [65,66,67]. A recent meta-analysis has demonstrated that large impairments in both phonological and spatial-visual domains of working memory are a characteristic of individuals with ASD independent of their IQ [1]. Spatial memory impairment has been studied previously in VPA rodent autistic models, mostly using the Morris water maze [68,69,70] or radial arm maze [71], and with some exception [72], learning and memory impairment for VPA models were highlighted. Our results are in line with these and also confirm a recent study using Korean ginseng to reverse autistic deficits in a VPA rat model that exhibited altered short-term memory [73]. We also have found that IN and less IP administered OXT improved short term memory in the Y maze, and that was correlated with a decline of oxidative stress status. OXT receptors are distributed abundantly in hippocampus and other brain regions involved in memory functions, suggesting OXT role in memory formation [74] and activation of hippocampal OXT receptors was showed to favor synaptic plasticity and transmission and suppressing neuronal background firing [75,76,77]. Several animals studies using IN OXT treatment in rodents with impaired hippocampal functioning or impaired memory have reported also positive effects [50,75,78,79]. More recently, it was found that OXT counteracts memory impairments on a cellular level by increasing hippocampal cyclic AMP–responsive element binding protein (CREB) phosphorylation [80] and by protecting the hippocampal extracellular signal-regulated kinase (ERK) signaling, a critically important pathway for synaptic plasticity and memory [50]. Abnormal ERK signaling may relate to sustained oxidative stress, a constant occurrence in ASD [21,22,23,24,25,26,27], which is known to alter phosphorylation processes [81]. Taken together, these data confirm our results indicating that IN OXT ameliorative effects on cognitive processes have an anti-oxidant component.”Anxiety and depression have been estimated as most prevalent among comorbidities commonly reported in ASD [82] and, similarly, the VPA rodent model has been validated by numerous studies to exhibit increased anxiety-like behavior [83] accompanied by heightened reactivity of the amygdala [84] and depressive-like manifestations [85]. Our study showed that OXT influences mood states and IN OXT exerts slightly higher anxiolytic effects and higher antidepressive effects when compared to IP. Similar to our results, IN OXT was reported to significantly reduce self-grooming and anxiety levels in a VPA-induced rat model of autism [38] and central (intraventricular injection) and not peripheral (intravenous injection) administered OXT was showed to alleviate symptoms in a behavioral despair rat model of depression [86]. Anxiolytic effect of OXT may lie in its inhibitory overall effect on amygdala activity [87], given the paraventricular oxytocinergic neurons dense projections to the hippocampus and the amygdala [88]. Two studies in mice that were either socially isolated or with deleted OXT-gene, reported significant down-regulation of OXTR in the central amygdala or increased immunoreactivity in amygdala and subsequent exacerbated anxiety-related behaviors [88,89]. OXT may act as antidepressant due to its inhibitory effect on the hypothalamic-pituitary-adrenal axis [90], the stress axis commonly reported overactive in depressed patients [91]. Thus, in a post-partum depression rat model induced by gestation restraint stress, local injection of OXT into the paraventricular nucleus reversed depressive-like behaviors and reduced also the high plasma corticosterone level [45].

Our results presented here are indicating that the positive improvement OXT brings in the cognitive and mood behavior of autistic rats may be associated with its effects on oxidative stress. Pearson correlations resulted in some significant correlations between behavioral parameters and oxidative markers suggesting an anti-oxidative effect of OXT. Another study of Wang et al. also reported that OXT treatment of prenatally VPA-exposed mice model similarly alleviated social and mood behaviors reducing elevated oxidative stress markers including GPx and MDA [92]. The mechanisms of OXT action at the molecular level regarding mood regulation remain unclear, but it is possible that the may be elicited by increased brain vulnerability to oxidative stress encountered in ASD [93].

The current results showed IN treatment had significantly better outcome than IP OXT in improving behavioral symptoms and central oxidative status. It would be obvious that central than peripheral administered OXT produces socio-emotional impact by reaching the behaviorally relevant brain areas, however only recently pharmacokinetics studies have brought evidence that nasal application permits direct delivery to the brain and [49,94]. Ref. [94] using liquid chromatography-mass spectrometry to analyze the disposition and absorption of OXT in rats brain found much higher concentration of OXT following IN compared to intravenous application.

OXT direct pathways to the CNS following IN administration include two primary means via olfactory and trigeminal nerve fibers [95]. Both these two ways imply delivery from the nasal cavity across the nasal epithelium through internalization of peptide followed by axonal transport and exocytosis or by extracellular diffusion or extracellular convection (bulk flow) [49,94,96]. The exact mechanisms remain yet unknown, as the hours-long endocytotic axonal transport may not allow OXT to survive internalization [96], whereas the polar structure of Oxt may cause low membrane permeability [49]. A series of earlier investigations showed that IN OXT led to increased OXT concentrations at the level of CNS after nasal administration, in rhesus macaques [97], and in the amygdala and hippocampus in rats and mice [98]. However, a space to interpretations remained on whether actually synthetic or endogenous OXT constituted these increases [98], as nasal administration of OXT may indirectly generate the release endogenous OXT by stimulating hypothalamic OXT autoreceptors in a positive feedback loop [99]. Only recently, by using OXT knockout mice [49] showed in for the first time that IN administration of OXT permeates the extracellular fluid of specific areas of the brain, i.e., the amygdala and hippocampus, where it peaks between 30-60 min [49].

In our study we observed that IP OXT also induced improvements in behavioral state and reduced to a certain extent the oxidative stress. Interestingly, in the same study of Smith et al. [49], IP administration of OXT was also followed by increases in the OXT concentrations in the brain, though less significant with a shortened return to baseline when compared to the IN route.

We may presume this to be related to a potential action on the brain, either directly, which implies crossing of the blood-brain barrier (BBB) or indirectly. According to some studies, the central effects of IP OXT might be mediated through the OXTR, widely distributed in the periphery at the level of reproductive organs, heart or gastrointestinal tract, that activate the vagal afferent pathways and send signals to the brain [96,99]. Ferris et al., using special fMRI imaging and computational analysis to follow signals of central and peripheral OXT along integrated neural circuits in rats, found that IP administered OXT does not cross the BBB, yet triggers OXT neurotransmission in the cerebellum and several brainstem areas through sensory visceral afferents [100]. Also, a study in mice showed that the anxiolytic-like effects of peripherally administered OXT were blocked by a centrally administered OXTR antagonist which does not cross the BBB [101].

Regarding the issue of crossing the BBB, it is well-known that OXT is a relatively large neuropeptide (1008 Da) which seriously limits its passage ability [96]. However, only recently, Yamamoto et al. identified an uptake molecular mechanism for OXT to cross the BBB, represented by a member of the immunoglobulin receptors class, the receptor for advanced glycation end-products (RAGE) on brain capillary endothelial cells [102]. Direct OXT-RAGE binding was confirmed using multiple methods including surface plasmon resonance and gel permeation chromatography [102]. Cerebrospinal fluid (CSF) measures and microperfusion results showed that the OXT increases in the mice brain which follow exogenous (subcutaneous, intravenous or IN) administration were lost after RAGE knockdown. An interesting hypothesis of Correia Lima and Rodrigues states that OXT may enter the CNS at the level of several circumventricular organs, such as the vascular organ of lamina terminalis or neurohypophysis, that lack the tight junctions between the capillaries cells [103].

Similarly to the many findings showing GI dysfunctions in subsets of autistic individuals such as chronic constipation, diarrhea, and abdominal pain [104], two studies have found changed and reduced richness of gut microbial composition [53] and histological abnormalities and decreased motility in GI tract [52] in rat offspring prenatally exposed to VPA. Given the significance of OXT for autistic deficits, we followed OXT effects on intestinal peristaltism and gastric emptying in our VPA rat model of autism. In line with the studies of Liu et al. [53] and Kim et al. [52] our data suggest VPA exposure to affect intestinal transit and cause constipation, whereas OXT treatment seems to balance bowel movements and fecal discharge. Only very recently, a study in chronic stress exposed mice with overreactive HPA axis and impaired gastric motility and gastric emptying reported that IN OXT treatment repaired the gastric symptoms and significantly reduced corticotropin-releasing factor mRNA expression and the corticosterone concentration [105]. These data are similar to our results suggesting that an ameliorating centrally mediated effect on intestinal transit by OXT. We have also observed an ameliorating effect of IP administered OXT. Despite paucity of data, OXT and OXT receptors were discovered to be expressed in the myenteric and submucous ganglia and nerve fibers throughout the entire gastrointestinal tract [106] and to influence important physiological responses, such as controlling GI transit, maintaining the intestinal mucosa or reducing intestinal inflammatory stress [107].

However, it is unclear whether OXT stimulates GI transit [107,108], or slows it—as observed in patients with diabetes mellitus and gastroparesis after intravenous infusions [109]. To our knowledge up to date there is no original study on the link between OXT and abnormalities of GI system in autism, however a plausible hypothesis was postulated by Welch and Klein [44] who explained the pathogenesis of autism as the result of abnormal gut-brain signaling pathways ultimately stemming from disrupted secretion of OXT hormone in the newborn. Thus, compromised endogenous fetal OT/OTR signaling during episodes of perinatal hypoxia would leave unchecked oxidative stress inducing a cascade of abnormal molecular reactions in the enteric neurons that would culminate in excessive afferent stimulation of the brain areas abnormal in autism [44].

## 5. Limitations of the Study

Among the limitations of the present study, we must mention the lack of control experiments to test the effects of IN and IP OXT in rats prenatally exposed to saline, although behavioral effects of IP oxytocin in rats have been previously studied by our group [45]; the reduced sample size of 5 rats which is at the inferior limit to permit drawing up relevant conclusions may also be considered a limitation. 

## 6. Conclusions

The present study evaluated the therapeutic efficacy of intranasal and intraperitoneal administrated OXT in early adulthood on the cognitive and mood deficits and elevated oxidative stress occurring after VPA prenatal exposure in rats. Whereas OXT alleviated behavioral and oxidative stress status in both routes of administration, intranasal treatment had significantly better outcome in improving short-term memory, reducing depressive manifestations and mitigating lipid peroxidation in the temporal lobes. Our results indicate that the OXT positive improvement in the cognitive and mood behavior of autistic rats may be associated with its effects on oxidative stress, as suggested by the significant correlations between these behavioral parameters and the specific markers of the oxidative stress status. Furthermore, a brief quantitative evaluation of the GI transit by fecal pellet count, suggested that OXT may have a balancing effect on the ccompromised gastrointestinal motility in the prenatally VPA-exposed rat model of autism.

## Figures and Tables

**Figure 1 medicina-56-00267-f001:**
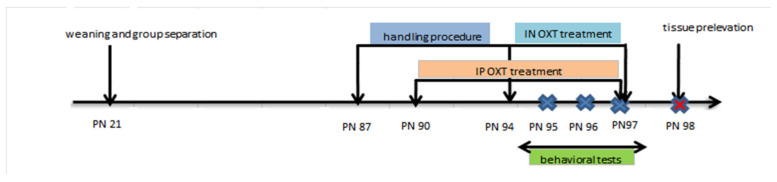
Protocol scheme for oxytocin intraperitoneal (IP) and intranasal (IN) treatment. PN: postnatal day; OXT: The hormone oxytocin.

**Figure 2 medicina-56-00267-f002:**
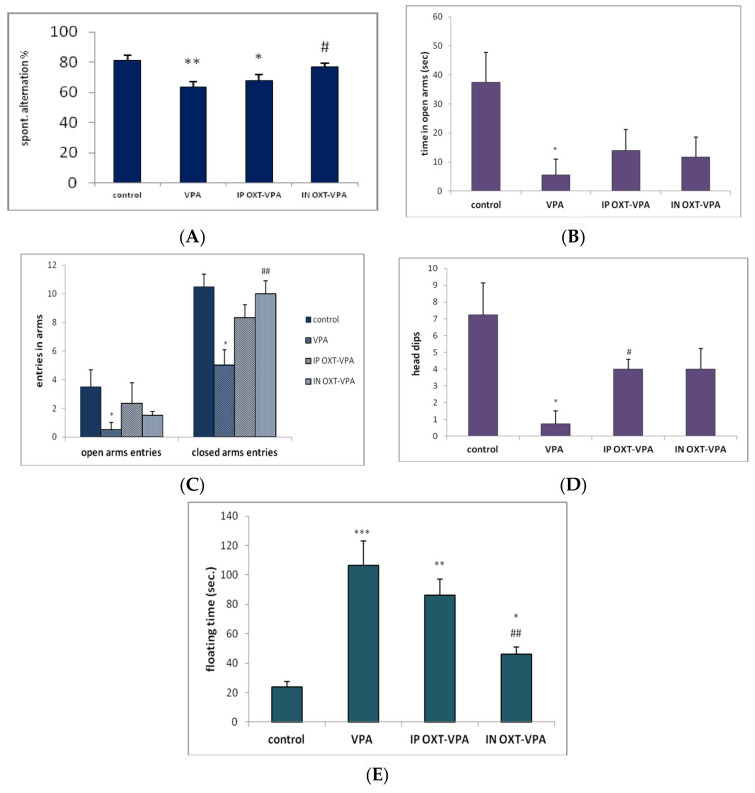
Behavioral effects of intraperitoneally and intranasally administrated OXT in prenatally VPA-exposed rat. The results are displayed as means ± SEM for *n* = 5 rats for each experimental group, * *p* < 0.05, ** *p* < 0.01 vs. control, *** *p* < 0.001 vs. control, # *p* < 0.05 vs. VPA group, ## *p* < 0.01 vs. VPA group. (**A**) OXT intranasal treatment shows better outcome than OXT intraperitoneally administration and improves the spontaneous alternation performance in prenatally VPA-exposed rats in the Y-maze task; (**B**) OXT (intranasal and intraperitoneal) administration increases time spent in open arms in prenatally VPA-exposed rats in the elevated plus maze task; (**C**) OXT (intranasal and intraperitoneal) administration increases number of entrances in the opened and closed arms of the EPM time spent in open arms in prenatally VPA-exposed rats in the elevated plus maze task; (**D**) OXT (intranasal and intraperitoneal) administration increases exploratory behavior in the EPM time as reflected by number of head dipping in prenatally VPA-exposed rats; (**E**) OXT intranasal alleviates depressive manifestations and decreases floating time in prenatally VPA-exposed rats.

**Figure 3 medicina-56-00267-f003:**
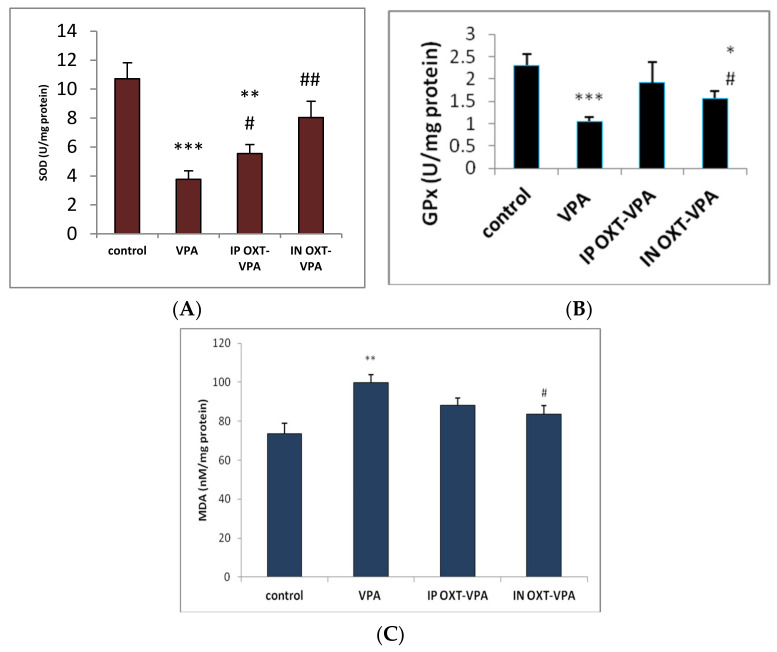
Effects of intraperitoneally and intranasally administrated OXT on oxidative stress biomarkers in prenatally VPA-exposed rat. The results are displayed as means ± SEM for *n* = 5 rats for each experimental group, * *p* < 0.05, ** *p* < 0.01 vs. control, *** *p* < 0.001 vs. control, # *p* < 0.05 vs. VPA group, ## *p* < 0.01 vs. VPA group. (**A**) Effects of intraperitoneally and intranasally administrated OXT on SOD activity in prenatally VPA-exposed rat temporal lobe homogenates; (**B**) Effects of intraperitoneally and intranasally administrated OXT on GPX activity in prenatally VPA-exposed rat temporal lobe homogenates; (**C**) Effects of intraperitoneally and intranasally administrated OXT on MDA levels in prenatally VPA-exposed rat temporal lobe homogenates.

**Table 1 medicina-56-00267-t001:** Correlations matrix of Pearson Correlations Coefficients.

		Spontaneous Alternation	Mobility	Open Arms Time	Closed Arms Entry	Head Dipping	MDA	SOD	GPx
Spontaneous alternation	Pearson Correlation	1	0.506 *	0.455 *	0.591 **	0.191	−0.727 **	0.538 *	0.660 **
	Sig. (2-tailed)		0.023	0.044	0.006	0.420	0.000	0.014	0.002
	N	20	20	20	20	20	20	20	20
Mobility	Pearson Correlation	0.506 *	1	0.514 *	0.228	0.262	−0.449 *	0.750 **	0.446 *
	Sig. (2-tailed)	0.023		0.020	0.334	0.264	0.047	0.000	0.049
	N	20	20	20	20	20	20	20	20
Open Arms time	Pearson Correlation	0.455 *	0.514 *	1	0.160	0.470 *	−0.387	0.504 *	0.288
	Sig. (2-tailed)	0.044	0.020		0.499	0.037	0.092	0.023	0.219
	N	20	20	20	20	20	20	20	20
Closed Arms entry	Pearson Correlation	0.591 **	0.228	0.160	1	0.450 *	−0.387	0.202	0.272
	Sig. (2-tailed)	0.006	0.334	0.499		0.046	0.092	0.393	0.246
	N	20	20	20	20	20	20	20	20
Head Dipping	Pearson Correlation	0.191	0.262	0.470 *	0.450 *	1	−0.090	0.139	−0.088
	Sig. (2-tailed)	0.420	0.264	0.037	0.046		0.706	0.560	0.712
	N	20	20	20	20	20	20	20	20
Malondialdehyde (MDA)	Pearson Correlation	−0.727 **	−0.449 *	−0.387	−0.387	−0.090	1	−0.539 *	−0.622 **
	Sig. (2-tailed)	0.000	0.047	0.092	0.092	0.706		0.014	0.003
	N	20	20	20	20	20	20	20	20
Superoxide dismutase (SOD)	Pearson Correlation	0.538 *	0.750 **	0.504 *	0.202	0.139	−0.539 *	1	0.460 *
	Sig. (2-tailed)	0.014	0.000	0.023	0.393	0.560	0.014		0.041
	N	20	20	20	20	20	20	20	20
Glutathione peroxidase (GPx)	Pearson Correlation	0.660 **	0.446 *	0.288	0.272	−0.088	−0.622 **	0.460 *	1
	Sig. (2-tailed)	0.002	0.049	0.219	0.246	0.712	0.003	0.041	
	N	20	20	20	20	20	20	20	20

* Correlation is significant at the 0.05 level (2-tailed). ** Correlation is significant at the 0.01 level (2-tailed).

**Table 2 medicina-56-00267-t002:** Simple regression analysis of the statistically significant correlations.

Response Variable (Dependent Variable Y)	Model Summary	Factor (Independent Variable X)							
		Spontaneous Alternation	Mobility	Open Arms Time	Closed Arms Time	Head Dipping	MDA	SOD	GPx
Spontaneous Alternation	Unstandardized Coefficients	-----------	Y = 50.584 + 0.123 × X	Y = 68.277 + 0.234 * X	Y = 55.730 + 1.964 × X		Y = 112.947 − 0.471 × X	Y = 62.768 + 1.375 × X	Y = 59.614 + 7.407 × X
	Sig. (Coefficients)	-----------	0.000 0.023	0.000 0.044	0.000 0.006		0.000 0.000	0.000 0.014	0.000 0.002
	R Square	-----------	0.256	0.207	0.349		0.529	0.290	0.435
Mobility	Unstandardized Coefficients	Y = 25.942 + 2.080 × X	------------	Y = 157.612 + 1.088 × X			Y = 279.379 − 1.149 × X	Y = 121.687 + 7.874 × X	Y = 141.052 + 20.577 × X
	Sig. (Coefficients)	0.675 0.023	------------	0.000 0.020			0.000 0.047	0.000 0.000	0.000 0.049
	R Square	0.256	------------	0.256			0.201	0.563	0.199
Open Arms time	Unstandardized Coefficients	Y = −46.748 + 0.884 × X	Y = −25.686 + 0.243 × X	---------		Y = 7.788 + 2.507 × X		Y = −0.163 + 2.501 × X	
	Sig. (Coefficients)	0.133 0.044	0.152 0.020	---------		0.166 0.037		0.984 0.023	
	R Square	0.207	0.265	---------		0.221		0.254	
Closed Arms time	Unstandardized Coefficients	Y = −4.425 + 0.178 × X			---------	Y = 7.046 + 0.372 × X			
	Sig. (Coefficients)	0.302 0.006			---------	0.000 0.046			
	R Square	0.349			---------	0.203			
Head Dipping	Unstandardized Coefficients			Y = 2.236 + 0.088 × X	Y = −0.847+ 0.545 × X	---------			
	Sig. (Coefficients)			0.027 0.037	0.710 0.046	---------			
	R Square			0.221	0.203	---------			
MDA	Unstandardized Coefficients	Y = 167.575 − 1.123 × X	Y = 116.106 − 0.169 × X				---------	Y = 101.098 − 2.126 × X	Y = 104.843 − 10.793 × X
	Sig. (Coefficients)	0.000 0.000	0.000 0.047				---------	0.000 0.014	0.000 0.003
	R Square	0.529	0.201				---------	0.290	0.387
SOD	Unstandardized Coefficients	Y = −8.300 + 0.211 × X	Y = −5.669 + 0.071 × X	Y = 5.191 + 0.102 × X			Y = 18.725 − 0.137 × X	---------	Y = 3.473 + 2.022 × X
	Sig. (Coefficients)	0.160 0.014	0.048 0.000	0.000 0.023			0.000 0.014	---------	0.059 0.041
	R Square	0.290	0.563	0.245			0.290	---------	0.212
GPx	Unstandardized Coefficients	Y = −2.536 + 0.059 × X	Y = 0.008 + 0.010 × X				Y = 4.812 −0.036 × X	Y = 0.987 + 0.105 × X	---------
	Sig. (Coefficients)	0.040 0.002	0.992 0.049				0.000 0.003	0.015 0.041	---------
	R Square	0.435	0.199				0.387	0.212	---------

## Data Availability

All of the data generated and analyzed during this study are included in this published article.

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
