# Peer review of "Oxytocin Differentiated Effects According to the Administration Route in a Prenatal Valproic Acid-Induced Rat Model of Autism"

_medicina, 2020, doi:10.3390/medicina56060267_

Round 1
Reviewer 1 Report
This is a review of “Oxytocin differentiated effects according to the administration route in a prenatal valproic acid-induced rat model of autism” by Lefter et al. This manuscript examines the different effects on behavioral and oxidative stress molecules in mice prenatally exposed to VPA and then administered oxytocin subchronically via intranasal or intraperitoneal injection. The question of the effectiveness between oral and peripheral administration of oxytocin is interesting. The data support the hypothesis that central administration of oxytocin is more effective than peripheral using tests for behaviors that are often co-morbid in people with ASD, though the differences were often small between the two routes of administration. Unfortunately, several technical details of the experiments preclude strong conclusions from the data.
The administration of saline or VPA at G12.5 is well validated in the literature. The authors do not note how the pups from each litter were allocated for the experiments which is very important when a maternal intervention has been used. Effectively, each full litter from any given dam has the same treatment and so all pups from the same litter in any treatment group should be averaged for a single representation. As there were only 3 saline treated dams, that would lower the control group to 3, instead of 5.
The sample size of 5 rats per group for behavioral tests generally does not have enough power to produce significant results. While it may have worked in the current experiments, the likelihood of replication is very small.
No females were used in the experiments. ASD used to be considered a disorder that affected males more than females, but it is becoming clear that females with ASD have a different behavioral manifestation than males and should always be included when potential treatments are being examined.
The IN and IP treatment goups are both compared against the VPA group which was administered saline IP in the same protocol as the IP-OXT group. The IP injections were done for 8 days with experiments starting on the 4th day, while the IN-OXT treatments began 1 day before behavioral tests. Therefore, IP-saline mice really aren’t a control for the IN-OXT mice.
The correlation data should be presented in a Table for ease of readability and non-significant correlations should also be presented.
The discussion is really long and doesn’t seem relevant. Not enough of the discussion addresses the hypothesis tested, the difference between central and peripheral administration. Specifically paragraphs 6-8 of the discussion need to be re-written so it is more clear how they relate to the current hypothesis and data. The random underlining in the discussion was also distracting and not useful for the reader.
Author Response
Reviewer 1
Comment 1. Oxytocin differentiated effects according to the administration route in a prenatal valproic acid-induced rat model of autism” by Lefter et al. This manuscript examines the different effects on behavioral and oxidative stress molecules in mice prenatally exposed to VPA and then administered oxytocin subchronically via intranasal or intraperitoneal injection. The question of the effectiveness between oral and peripheral administration of oxytocin is interesting. The data support the hypothesis that central administration of oxytocin is more effective than peripheral using tests for behaviors that are often co-morbid in people with ASD, though the differences were often small between the two routes of administration. Unfortunately, several technical details of the experiments preclude strong conclusions from the data.
The administration of saline or VPA at G12.5 is well validated in the literature. The authors do not note how the pups from each litter were allocated for the experiment which is very important when a maternal intervention has been used. Effectively, each full litter from any given dam has the same treatment and so all pups from the same litter in any treatment group should be averaged for a single representation. As there were only 3 saline treated dams, that would lower the control group to 3, instead of 5.
Answer: Currently, the effect of within-litter similarity has been discussed and is in most cases commonly acknowledged (1). However, studies are reporting within-litter variances in rats maternal behavior that may relate to inter individual offspring behavioral variance (2,3). In the current experiment considering resource shortage and on ethical grounds, the option was to use multiple animals per litter, conferring the experiment a mixture of a between-litter and a within-litter design (4), which is useful to limit the within-litter effects. 1. Lazic, S. E., Essioux, L. (2013). Improving basic and translational science by accounting for litter-to-litter variation in animal models. BMC neuroscience, 14, 372. Cavigelli SA, Ragan CM, Barrett CE, Michael KC. Within-litter variance in rat maternal behaviour. Behav Processes. 2010 Jul;84(3):696-704.3. van Hasselt FN, Tieskens JM, Trezza V, Krugers HJ, Vanderschuren LJ, Joëls M. Within-litter variation in maternal care received by individual pups correlates with adolescent social play behavior in male rats. Physiol Behav. 2012 Jul 16;106(5):701-6 4. Festing MF. Design and statistical methods in studies using animal models of development. ILAR J. 2006;47(1):5-14. Review
Comment 2. The sample size of 5 rats per group for behavioral tests generally does not have enough power to produce significant results. While it may have worked in the current experiments, the likelihood of replication is very small.
Answer: We are well aware of the small sample sized groups used in the current experiment, and consider this aspect as a limitation - that was due, as in many other studies, both to ethical concerns and financial constraints - and mentioned at the end of the current work. We considered the number of animals used in this study to be enough to investigate whether differences appear between two ways of administration, and in the end we were able to identify some significant points in this regard, which by itself justifies to analogous proportion the design power – indeed it would seem that the current sample size may represent the inferior limit to permit drawing up relevant conclusions. There is no general rule in using a certain number of animals to prove a hypothesis, authors have discussed this issue and we can cite Anderson and Vingrys (1) arguing that “despite criticisms a sample size of five may well be useful in scientific research, although to provide more confident estimates of the population proportion, much larger numbers are needed.” There are studies using small sample sizes in experiments to investigate different aspects that are well accepted by scientific community (2-4). Regarding the calculation of the sample size, one way to approach this is the mathematical power analysis method based on a series of variables, however this would have required for a biostatistician specialist, considering the more complex character of the experiment. Instead, our calculation was performed and cofirmed by the resource equation method, an easier method to employ, though considered somewhat simplistic, but valid nonetheless and a useful addition to ‘‘common sense’’(5).
- Andrew John Anderson, Algis Jonas Vingrys; Small Samples: Does Size Matter?. Invest. Ophthalmol. Vis. Sci.2001;42(7):1411-1413. doi:https://doi.org/.
- R. J. Windle, N. Shanks, S. L. Lightman, C. D. Ingram, Central Oxytocin Administration Reduces Stress-Induced Corticosterone Release and Anxiety Behavior in Rats, Endocrinology, Volume 138, Issue 7, 1 July 1997, Pages 2829–2834, https://doi.org/10.1210/endo.138.7.5255
3. Mizuno A, Cherepanov SM, Kikuchi Y, Fakhrul AA, Akther S, Deguchi K, Yoshihara T, Ishihara K, Shuto S, Higashida H. Lipo-oxytocin-1, a Novel Oxytocin Analog Conjugated with Two Palmitoyl Groups, Has Long-Lasting Effects on Anxiety-Related Behavior and Social Avoidance in CD157 Knockout Mice. Brain Sci. 2015 Jan 20;5(1):3-13. 4. Peñagarikano O, Lázaro MT, Lu XH, Gordon A, Dong H, Lam HA, Peles E, Maidment NT, Murphy NP, Yang XW, Golshani P, Geschwind DH. Exogenous and evoked oxytocin restores social behavior in the Cntnap2 mouse model of autism. Sci Transl Med. 2015 Jan 21;7(271):271ra8. 5. Festing, M. F. (2018). On determining sample size in experiments involving laboratory animals. Laboratory Animals, 52(4), 341–350.
Comment 3. No females were used in the experiments. ASD used to be considered a disorder that affected males more than females, but it is becoming clear that females with ASD have a different behavioral manifestation than males and should always be included when potential treatments are being examined.
Answer: It is a recent tendency to consider both sexes in animals studies when studying ASD, to which we agree; however there must be a serious reason for ASD manifestations to be “less penetrant” (1) in females than males in human cases, speculated to be, rather than missed diagnosis in females, a molecular protective system against excess mutational burden (2, as reviewed by 1). The animal models in our study were intentioned to represent autistic-like features by exposure to VPA. In this regard, when it comes to prenatal VPA-exposure as etiologic factor for ASD, the study of sex differences indicates female rats/rodents to be “less vulnerable to the deleterious effects of prenatal VPA exposure on social communication, emotional reactivity and cognitive performance than male rats” (3). Sexual dimorphism in behavioral, and also genetic, neuroendocrine and immune profiles clearly showing male animals (and humans) prone to greater impairment, largely reviewed by (1), should be illustrative that at least in the particular case of VPA, focusing on males is well justified. 1. Jeon, S. J., Gonzales, E. L., Mabunga, D., Valencia, S. T., Kim, D. G., Kim, Y., Adil, K., Shin, D., Park, D., & Shin, C. Y. (2018). Sex-specific Behavioral Features of Rodent Models of Autism Spectrum Disorder. Experimental neurobiology, 27(5), 321–343. https://doi.org/10.5607/en.2018.27.5.321 2. Jacquemont S, Coe BP, Hersch M, Duyzend MH, Krumm N, Bergmann S, Beckmann JS, Rosenfeld JA, Eichler EE. A higher mutational burden in females supports a "female protective model" in neurodevelopmental disorders. Am J Hum Genet. 2014 Mar 6;94(3):415-25.3. Melancia F, Schiavi S, Servadio M, et al. Sex-specific autistic endophenotypes induced by prenatal exposure to valproic acid involve anandamide signalling. British Journal of Pharmacology. 2018 Sep;175(18):3699-3712.
Comment 4. The IN and IP treatment goups are both compared against the VPA group which was administered saline IP in the same protocol as the IP-OXT group. The IP injections were done for 8 days with experiments starting on the 4th day, while the IN-OXT treatments began 1 day before behavioral tests. Therefore, IP-saline mice really aren’t a control for the IN-OXT mice.
Answer: This is indeed a valid observation, which we may consider it one of the serious limitations of the current study, if not for some aspects. In this regard, we can nominate: firstly, IN treatment protocol was performed for a longer period of time, as it was initiated with one week prior to the beginning of the tests, which, considering the entire procedure, would constitute a significant load of stress. The reason to administrate saline by IP or IN path is to expose to a similar amount of stress the groups of rats. Secondly, the main objective in this study was to compare IN administration of OXT specifically with IP administration, not with naïve control groups. The fact that IN administration is considered a non-invasive administration route that also can attenuate stress response (1) is further positive elements in favor of IN path.1. Yang, Y., Yu, H., Babygirija, R., Shi, B., Sun, W., Zheng, X., & Zheng, J. (2019). Intranasal Administration of Oxytocin Attenuates Stress Responses Following Chronic Complicated Stress in Rats. Journal of neurogastroenterology and motility, 25(4), 611–622. https://doi.org/10.5056/jnm19065
Comment 5. The correlation data should be presented in a Table for ease of readability and non-significant correlations should also be presented.
Answer: this has been addressed by adding the tables in the text.
Comment 6. The discussion is really long and doesn’t seem relevant. Not enough of the discussion addresses the hypothesis tested, the difference between central and peripheral administration. Specifically paragraphs 6-8 of the discussion need to be re-written so it is clearer how they relate to the current hypothesis and data. The random underlining in the discussion was also distracting and not useful for the reader.
Answer: thank you for these indications; we have re-written the specificated paragraphs (see below) and reshaped the Discussions into a hopefully more coherent construct.

Reviewer 2 Report
Comments to Author
In their article, Lefter et al. investigated the differentiated effects oxytocin enfolds according to the administration route rat model of autism. To investigate this, the authors used a prenatal valproic acid-induced rat model and administered oxytocin either intranasally (IN) or intraperitoneally (IP). Their main findings can be summarized as follows: oxytocin improved behavioral as well as oxidative stress regardless of how oxytocin was administered. However IN administration resulted in better outcomes. IN oxytocin led to improved short-term memory and alleviated depressive manifestations, it further reduced lipid peroxiation in the temporal lobes.
Introduction:
-line 53: “Different comorbidities occur often along with the core symptoms [7], such as gastrointestinal disorders...” to my knowledge there are theories about causal relationships of GI bacteria and the development of autism, maybe this is worth looking into this
-line 84: please explain abbreviations before use
-I find it surprising that most of the literature cited in the introduction refers to human research
-line 101: “Intranasal (IN) administration seems to be more suitable for the study of neuropsychiatric disorders, as it allows the neuropeptide to be absorbed through the nasal mucosa and cross the blood-brain barrier.” I would be interestred to see the reference for this statement. As far as I know this is only one assumption of how IN enters the brain/ CSF. Please see: “The promise and pitfalls of intranasally administering psychopharmacological agents for the treatment of psychiatric disorders.”
method & results
-I am not an animal study expert, so please excuse if my comments are not helpful, but what I stumbled upon was the following:
-small sample size: in total 20 (5 in each of the 4 groups). Is this common for these kind of studies? Have you done a power analysis which justifies such a small sample, or have you calculated the observed power? If this is common practice, please just add this information, however it would be good to know whether such small samples can justify statements on outcomes.
-Was rat behavior tested, before oxtytocin was administred to varify that “autism induction” worked in the first place? Or if this is unusual, please explain why or give other study examples.
-Many correlations are reported, I am wondering whether it has been corrected for multiple comparisons, or whether each correlation was calculated separately?
Discussion
-A large part of the discussion deals with oxytocin effects on memory. However, it is not entirely clear to me how memory deficits are related to autism. In general the relationship between the topics might be integrated a bit more to facilitate the reading. In the past OXT’s effects on memory per se (at least in human research) have been found to impair memory (in various ways), please see Heinrichs et al. 2004: Selective amnesic effects of oxytocin on human memory. Physiol Behav, 83(1), 31-38.
More recently, OXT was found to positively influence memory (in animal studies), showing that oxytocin counteracts memory impairments caused by stress on a cellular level, thereby preventing memory impairments, see Lee et al. 2015: Oxytocin Protects Hippocampal Memory and Plasticity from Uncontrollable Stress
While reading the part of the discussion concerning memory, I was wondering how this is related to autism? Is memory used as an example for cognitive improvement? And is there literature on autism showing that memory is impaired in autism, especially in the domains which are discussed here?
-line 429 please explain abbreviation before use
-the paragrpah starting at line 484 “Our results presented here are indicating that the OXT positive improvement in the cognitive and mood behavior of autistic rats brings may be associated with its effects on oxidative stress.” is a good example of what I would expect more in the discussion. It nicely integraties the reported results into the already existing findings and bigger context.
-line 517: “To our knowledge up to date there is no original study on the link between OXT and abnormalities of GI system...” as stated above, I think there is literature on this topic. As a start you might look into this: “Association Between Gut Microbiota and Autism Spectrum Disorder: A Systematic Review and Meta-Analysis”
General
The study is interesting and the paper is well written. In my opinion, it could be emphasized a little bit more which administration route is favourable to induce oxytocin effects, or why such a general statement cannot be made. And generally the information provided in this article could be integrated more so that the reader can follow the authors’ thoughts a bit better. As stated above, one paragraph in the discussion has succeeded well in doing so.
Author Response
Reviewer 2
Comment 1. In their article, Lefter et al. investigated the differentiated effects oxytocin enfolds according to the administration route rat model of autism. To investigate this, the authors used a prenatal valproic acid-induced rat model and administered oxytocin either intranasally (IN) or intraperitoneally (IP). Their main findings can be summarized as follows: oxytocin improved behavioral as well as oxidative stress
regardless of how oxytocin was administered. However IN administration resulted in better outcomes. IN oxytocin led to improved short-term memory and alleviated depressive manifestations, it further reduced lipid peroxiation in the temporal lobes.
Introduction:
-line 53: “Different comorbidities occur often along with the core symptoms [7], such as gastrointestinal disorders...” to my knowledge there are theories about causal relationships of GI bacteria and the development of autism, maybe this is worth looking into this
Answer: we have introduced in text a fragment on the potential causality between GI disturbances and ASD, with reference on animal studies, as follows:
Line 58-65: The correlations between GI symptoms and ASD behavioral phenotype carefully revised in a very systematical meta-analysis seem to relate to alterations of gut microbiota composition (Xu et al., 2019). Dinan and Cryan suggest microbiome to be implicated in ASD etiology and cite a series of studies on germ‐free animals that show the impact of microbiome during neurogenesis period on amygdala (LeDoux, 2007; Stilling et al., 2014). Microbial genes, brain & behaviour ‐ epigenetic regulation of the gut‐brain axis. Genes Brain Behav 13, 69–86.) and hippocampus (Ogbonnaya et al., 2015) critical brain areas for anxiety and fear‐related behavior and memory respectively (Dinan and Cryan, 2017).
- Xu, M., Xu, X., Li, J., & Li, F. (2019). Association Between Gut Microbiota and Autism Spectrum Disorder: A Systematic Review and Meta-Analysis. Frontiers in psychiatry, 10, 473. https://doi.org/10.3389/fpsyt.2019.00473
- LeDoux J (2007). The amygdala. Curr Biol 17, R868–874.
- Stilling RM, Dinan TG & Cryan JF (2014). Microbial genes, brain & behaviour ‐ epigenetic regulation of the gut‐brain axis. Genes Brain Behav 13, 69–86.)
- Ogbonnaya ES, Clarke G, Shanahan F, Dinan TG, Cryan JF & O'Leary OF (2015). Adult hippocampal neurogenesis is regulated by the microbiome. Biol Psychiatry 78, e7–9.
- Dinan, T.G. and Cryan, J.F. (2017), Gut instincts: microbiota as a key regulator of brain development, ageing and neurodegeneration. J Physiol, 595: 489-503. doi:10.1113/JP273106)
Comment 2. line 84: please explain abbreviations before use:
Answer: It has been corrected - reactive oxygen species added to” ROS” in text.
Comment 3. I find it surprising that most of the literature cited in the introduction refers to human research
Answer: We have tried to make the Introduction chapter as relevant as possible to ASD as a human disorder, so we included both human and animal studies. There are animal studies cited whenever the argumentative process made it necessary: references 12-15, 25; 36-41; 43, 44; furthermore we have added several references in the text:
Line 58-65:Ogbonnaya ES, Clarke G, Shanahan F, Dinan TG, Cryan JF & O'Leary OF (2015). Adult hippocampal neurogenesis is regulated by the microbiome. Biol Psychiatry 78, e7–9
LeDoux J (2007). The amygdala. Curr Biol 17, R868–874.;
Stilling RM, Dinan TG & Cryan JF (2014). Microbial genes, brain & behaviour ‐ epigenetic regulation of the gut‐brain axis. Genes Brain Behav 13, 69–86.)
Line 98-101:”Polymorphisms of genes encoding the proteins and related regulatory factors such as the OXT-NPI (encoding the OXT prepropeptide), OXTR or CD38 genes (involved in OXT release in the brain), are associated with social deficits and repetitive behavior and have been reported in autistic populations [28,29]. Similarly, findings in animal models relate impairments of the OXT system to ASD symptoms such as social memory (1) restricted interests to novelty (2) or social communication and interaction (3,4).”
- Aseel Yaseen, Kuldeep Shrivastava, Zohar Zuri, Ossama A Hatoum, Mouna Maroun, Prefrontal Oxytocin is Involved in Impairments in Prefrontal Plasticity and Social Memory Following Acute Exposure to High Fat Diet in Juvenile Animals, Cerebral Cortex, Volume 29, Issue 5, May 2019, Pages 1900–1909, https://doi.org/10.1093/cercor/bhy070
- Leonzino M, Ponzoni L, Braida D, Gigliucci V, Busnelli M, Ceresini I, Duque-Wilckens N, Nishimori K, Trainor BC, Sala M, Chini B. Impaired approach to novelty and striatal alterations in the oxytocin receptor deficient mouse model of autism. Horm Behav. 2019 Aug;114:104543. doi: 10.1016/j.yhbeh.2019.06.007
- Sala M, Braida D, Donzelli A, Martucci R, Busnelli M, Bulgheroni E, Rubino T, Parolaro D, Nishimori K, Chini B (2013) Mice heterozygous for the oxytocin receptor gene (Oxtr(+/)) show impaired social behaviour but not increased aggression or cognitive inflexibility: evidence of a selective haploinsufficiency gene effect. J Neuroendocrinol 25(2):107–118.
- Dai, Y. C., Zhang, H. F., Schön, M., Böckers, T. M., Han, S. P., Han, J. S., & Zhang, R. (2018). Neonatal Oxytocin Treatment Ameliorates Autistic-Like Behaviors and Oxytocin Deficiency in Valproic Acid-Induced Rat Model of Autism. Frontiers in cellular neuroscience, 12, 355. https://doi.org/10.3389/fncel.2018.00355
Comment 4. line 101: “Intranasal (IN) administration seems to be more suitable for the study of neuropsychiatric disorders, as it allows the neuropeptide to be absorbed through the nasal mucosa and cross the blood-brain barrier.” I would be interestred to see the reference for this statement. As far as I know this is only one assumption of how IN enters the brain/ CSF. Please see: “The promise and pitfalls of intranasally administering psychopharmacological agents for the treatment of psychiatric disorders.”
Answer: We thank you for this observation. Indeed, we have found a recent 2019 study to take a further step in bringing clearer evidence to the direct transport of OXT to the brain following IN administration, alongside other previous works that also support this hypothesis (cited in the discussion such as [89]. Tanaka, A.; Furubayashi, T.; Arai, M.; Inoue, D.; Kimura, S.; Kiriyama, A.; Kusamori, K.; Katsumi, H.; Yutani, R.; Sakane, T.; Yamamoto, A. Delivery of Oxytocin to the Brain for the Treatment of Autism Spectrum Disorder by Nasal Application. Mol Pharm. 2018, 15, 1105-1111). We have changed the inaccurate phrase to the following and added the reference:
Lines 113-119 ”… the non-invasive intranasal (IN) administration may be a more efficient way in approaching central nervous system (CNS) diseases, as this route was recently reported to allow the neuropeptide to bypass the blood-brain barrier and permeate into specific areas of the brain” [1].
We have considered the suggested review for the Discussion chapter.
- Smith, A. S., Korgan, A. C., & Young, W. S. (2019). Oxytocin delivered nasally or intraperitoneally reaches the brain and plasma of normal and oxytocin knockout mice. Pharmacological Research, 104324. doi:10.1016/j.phrs.2019.104324
Comment 5. Method & results
- I am not an animal study expert, so please excuse if my comments are not helpful, but what I stumbled upon was the following:
- small sample size: in total 20 (5 in each of the 4 groups). Is this common for these kind of studies? Have you done a power analysis which justifies such a small sample, or whether such small samples can justify statements on outcomes. have you calculated the observed power? If this is common practice, please just add this information, however it would be good to know
Answer: as this very good question is similar to Reviewer 1’s comment 2, we kindly ask Reviewer 2 to consider our answer to the aforementioned comment:” We are well aware of the small sample sized groups used in the current experiment, and consider this aspect as a limitation - that was due, as in many other studies, both to ethical concerns and financial constraints - and mentioned at the end of the current work. We considered the number of animals used in this study to be enough to investigate whether differences appear between two ways of administration, and in the end we were able to identify some significant points in this regard, which by itself justifies to analogous proportion the design power – indeed it would seem that the current sample size may represent the inferior limit to permit drawing up relevant conclusions. There is no general rule in using a certain number of animals to prove a hypothesis, authors have discussed this issue and we can cite Anderson and Vingrys (1) arguing that “despite criticisms a sample size of five may well be useful in scientific research, although to provide more confident estimates of the population proportion, much larger numbers are needed.” There are studies using small sample sizes in experiments to investigate different aspects that are well accepted by scientific community (2-4). Regarding the calculation of the sample size, one way to approach this is the mathematical power analysis method based on a series of variables, however this would have required for a biostatistician specialist considering the more complex character of the experiment. Instead, our calculation was performed and confirmed by the resource equation method, easier though considered somewhat simplistic, but valid nonetheless and a useful addition to ‘‘common sense’’ (5).”
Comment 6. Was rat behavior tested, before oxtytocin was administred to varify that “autism induction” worked in the first place? Or if this is unusual, please explain why or give other study examples.
Answer: The methodolgy of inducing ASD-like symptomatology in rats used in the study is largely validated and accepted and we have verified it in other previous experiments that were published - references 44-46 - or others that data remain yet unpublished. However, this is indeed one of the study’s limitations.
Comment 7. Many correlations are reported, I am wondering whether it has been corrected for multiple comparisons, or whether each correlation was calculated separately?
Answer: Pearson correlations were calculated and also simple and multiple regression analysis was performed. We have added only Pearson correlations and simple regression tables in the text for use of readability.
Comment 8. Discussion
A large part of the discussion deals with oxytocin effects on memory. However, it is not entirely clear to me how memory deficits are related to autism. In general the relationship between the topics might be integrated a bit more to facilitate the reading. In the past OXT’s effects on memory per se (at least in human research) have been found to impair memory (in various ways), please see Heinrichs et al. 2004: Selective amnesic effects of oxytocin on human memory. Physiol Behav, 83(1), 31-38. More recently, OXT was found to positively influence memory (in animal studies), showing that oxytocin counteracts memory impairments caused by stress on a cellular level, thereby preventing memory impairments, see Lee et al. 2015: Oxytocin Protects Hippocampal Memory and Plasticity from Uncontrollable Stress. While reading the part of the discussion concerning memory, I was wondering how this is related to autism? Is memory used as an example for cognitive improvement? And is there literature on autism showing that memory is impaired in autism, especially in the domains which are discussed here?
Answer: We are referring to the short-term memory, which has indeed been referenced by a series of studies [54-56 in text] including a very recent meta-analysis to be highly relevant in ASD [Habib, A., Harris, L., Pollick, F., Melville, C. (2019). A meta-analysis of working memory in individuals with autism spectrum disorders. PloS one, 14(4), e0216198. https://doi.org/10.1371/journal.pone.0216198 ] as it ”concerns the cognitive domains involved in social impairments, communication problems, and repetitive activities” and is related to the different activation patterns of brain regions in ASD individuals as identified by fMRI [Habib et al.,2019]. We also have decided to shorten the too large section allocated to memory – which is why we will not present the more complex facets regarding the influence of OXT on memory that was found to have indeed a subjective component –context, personality - that determines the outcome on memory accuracy, whether positive or negative, as suggested in Heinrichs et al. 2004: Selective amnesic effects of oxytocin on human memory. Physiol Behav, 83(1), 31-38. (cf. Wagner, U., & Echterhoff, G. (2018). When Does Oxytocin Affect Human Memory Encoding? The Role of Social Context and Individual Attachment Style. Frontiers in human neuroscience, 12, 349. https://doi.org/10.3389/fnhum.2018.00349 ). We have included the suggested study of Lee et al., 2015.
“There is a growing view that working memory deficits, that involve cognitive domains involved in social impairments, communication problems, and repetitive activities [2 cited by 1], may be highly relevant in ASD [1]. Although not unanimously signaled in autistic individuals, majority of neuropsychological studies clearly indicate at least some degree of deficit in spatial and spatial-visual working memory tasks [54-56]. A recent meta-analysis has demonstrated that large impairments in both phonological and spatial-visual domains of working memory are a characteristic of individuals with ASD independent of their IQ [1]. Spatial memory impairment has been studied previously in VPA rodent autistic models, mostly using the Morris water maze [57-59] or radial arm maze [60], and with some exception [61], learning and memory impairment for VPA models were highlighted. Our results are in line with these and also confirm a recent study using Korean ginseng to reverse autistic deficits in a VPA rat model that exhibited altered short-term memory [62]. We also have found that IN and less IP administered OXT improved short term memory in the Y maze, and that was correlated with a decline of oxidative stress status. OXT receptors are distributed abundantly in hippocampus and other brain regions involved in memory functions, suggesting OXT role in memory formation [3] and activation of hippocampal OXT receptors was showed to favor synaptic plasticity and transmission and suppressing neuronal background firing [66,68,69]. Several animals studies using IN OXT treatment in rodents with impaired hippocampal functioning or impaired memory have reported also positive effects [41,66,67,72]. More recently, it was found that OXT counteracts memory impairments on a cellular level by increasing hippocampal cyclic AMP–responsive element binding protein (CREB) phosphorylation [71] and by protecting the hippocampal extracellular signal-regulated kinase (ERK) signaling, a critically important pathway for synaptic plasticity and memory [41]. Abnormal ERK signaling may relate to sustained oxidative stress, a constant occurrence in ASD [16-22], which is known to alter phosphorylation processes [4]. Taken together, these data confirm our results indicating that IN OXT ameliorative effects on cognitive processes have an anti-oxidant component.”
- Habib, A., Harris, L., Pollick, F., Melville, C. (2019). A meta-analysis of working memory in individuals with autism spectrum disorders. PloS one, 14(4), e0216198. https://doi.org/10.1371/journal.pone.0216198.
- Barendse EM, Hendriks MP, Jansen JF, Backes WH, Hofman PA, Thoonen G, Kessels RP, Aldenkamp AP. Working memory deficits in high-functioning adolescents with autism spectrum disorders: neuropsychological and neuroimaging correlates. J Neurodev Disord. 2013 Jun 4;5(1):14.
3. Lin YT, Hsieh TY, Tsai TC, Chen CC, Huang CC, Hsu KS. Conditional Deletion of Hippocampal CA2/CA3a Oxytocin Receptors Impairs the Persistence of Long-Term Social Recognition Memory in Mice. J Neurosci. 2018 Jan 31;38(5):1218-1231. 4. Zhang L, Jope RS. Oxidative stress differentially modulates phosphorylation of ERK, p38 and CREB induced by NGF or EGF in PC12 cells. Neurobiology of Aging. 1999 May-Jun;20(3):271-278. DOI: 10.1016/s0197-4580(99)00049-4.
Comment 9. Line 429 please explain abbreviation before use.
Answer: We have added explanation: hypothalamic pituitary adrenal (HPA).
Comment 10. The paragraph starting at line 484 “Our results presented here are indicating that the OXT positive improvement in the cognitive and mood behavior of autistic rats brings may be associated with its effects on oxidative stress.” is a good example of what I would expect more in the discussion. It nicely integrates the reported results into the already existing findings and bigger context.
Answer: Thank you for the suggestion, we have modified the Discussions chapter with the intention of making it more coherent.
Comment 11. Line 517: “To our knowledge up to date there is no original study on the link between OXT and abnormalities of GI system...” as stated above, I think there is literature on this topic. As a start you might look into this: “Association Between Gut Microbiota and Autism Spectrum Disorder: A Systematic Review and Meta-Analysis”
Answer: Thank you again for the reference; we have changed the phrasing and added the reference in the Introduction chapter; however, we have to reply that, as we are also aware that there are many studies on the ASD and GI disturbances, including disbyosis, we couldn’t find, up to the date when we wrote the current article, original studies on any potential mechanisms on the link between OXT and abnormalities of GI system in autism - which our results, limited to only a quantitative evaluation of fecal discharge, relate to - except for Dr. Welch’s study, that we have cited [35]. However, during this revision period, we have found another paper on “Intranasal Administration of Oxytocin Attenuates Stress Responses Following Chronic Complicated Stress in Rats” that we added in our current article, and from which we also cite: “however, the effect of intranasal administration of OXT on … GI motility have not been investigated”. [Yang, Y., Yu, H., Babygirija, R., Shi, B., Sun, W., Zheng, X., & Zheng, J. (2019). Intranasal Administration of Oxytocin Attenuates Stress Responses Following Chronic Complicated Stress in Rats. Journal of neurogastroenterology and motility, 25(4), 611–622. https://doi.org/10.5056/jnm19065 ].
Comment 13.
General
The study is interesting and the paper is well written. In my opinion, it could be emphasized a little bit more which administration route is favourable to induce oxytocin effects, or why such a general statement cannot be made. And generally the information provided in this article could be integrated more so that the reader can follow the authors’ thoughts a bit better. As stated above, one paragraph in the discussion has succeeded well in doing so.
Answer: Thank you for the suggestions. We have made some modifications to the Discussions considering these suggestions and also have detailed the part concerning the IN-IP route. Below is the modified version of the chapter until the GI section.
Anxiety and depression have been estimated as most prevalent among comorbidities commonly reported in ASD [75] and, similarly, the VPA rodent model has been validated by numerous studies to exhibit increased anxiety-like behavior [76] accompanied by heightened reactivity of the amygdala [77] and depressive-like manifestations [78]. Our study showed that OXT influences mood states and IN OXT exerts slightly higher anxiolytic effects and higher antidepressive effects when compared to IP. Similar to our results, IN OXT was reported to significantly reduce self-grooming and anxiety levels in a VPA-induced rat model of autism [36] and central (intraventricular injection) and not peripheral (intravenous injection) administered OXT was showed to alleviate symptoms in a behavioral despair rat model of depression [86]. Anxiolytic effect of OXT may lie in its inhibitory overall effect on amygdala activity [1], given the paraventricular oxytocinergic neurons dense projections to the hippocampus and the amygdala [83]. Two studies in mice that were either socially isolated or with deleted OXT-gene, reported significant down-regulation of OXTR in the central amygdala or increased immunoreactivity in amygdala and subsequent exacerbated anxiety-related behaviors [83,84]. OXT may act as antidepressant due to its inhibitory effect on the hypothalamic-pituitary-adrenal axis [85], the stress axis commonly reported overactive in depressed patients [2]. Thus, in a post-partum depression rat model induced by gestation restraint stress, local injection of OXT into the paraventricular nucleus reversed depressive-like behaviors and reduced also the high plasma corticosterone level [37].
Our results presented here are indicating that the positive improvement OXT brings in the cognitive and mood behavior of autistic rats may be associated with its effects on oxidative stress. Pearson correlations resulted in some significant correlations between behavioral parameters and oxidative markers suggesting an anti-oxidative effect of OXT. Another study of Wang et al. also reported that OXT treatment of prenatally VPA-exposed mice model similarly alleviated social and mood behaviors reducing elevated oxidative stress markers including GPx and MDA [37b]. The mechanisms of OXT action at the molecular level regarding mood regulation remain unclear, but it is possible that the may be elicited by increased brain vulnerability to oxidative stress encountered in ASD [93].
- Sobota, R., Mihara, T., Forrest, A., Featherstone, R. E., & Siegel, S. J. (2015). Oxytocin reduces amygdala activity, increases social interactions, and reduces anxiety-like behavior irrespective of NMDAR antagonism. Behavioral neuroscience, 129(4), 389–398. https://doi.org/10.1037/bne0000074
- 2. Varghese, F. P., & Brown, E. S. (2001). The Hypothalamic-Pituitary-Adrenal Axis in Major Depressive Disorder: A Brief Primer for Primary Care Physicians. Primary care companion to the Journal of clinical psychiatry, 3(4), 151–155. https://doi.org/10.4088/pcc.v03n0401
37b. Wang, Y., Zhao, S., Liu, X., Zheng, Y., Li, L., & Meng, S. (2018). Oxytocin improves animal behaviors and ameliorates oxidative stress and inflammation in autistic mice. Biomedicine & Pharmacotherapy, 107, 262–269. doi:10.1016/j.biopha.2018.07.148
The current results showed IN treatment had significantly better outcome than IP OXT in improving behavioral symptoms and central oxidative status. It would be obvious that central than peripheral administered OXT produces socio-emotional impact by reaching the behaviorally relevant brain areas, however only recently pharmacokinetics studies have brought evidence that nasal application permits direct delivery to the brain and [89,1]. [89] using liquid chromatography-mass spectrometry to analyze the disposition and absorption of OXT in rats brain found much higher concentration of OXT following IN compared to intravenous application.
OXT direct pathways to the CNS following IN administration include two primary means via olfactory and trigeminal nerve fibers [2]. Both these two ways imply delivery from the nasal cavity across the nasal epithelium through internalization of peptide followed by axonal transport and exocytosis or by extracellular diffusion or extracellular convection (bulk flow) [89,3,1]. The exact mechanisms remain yet unknown, as the hours-long endocytotic axonal transport may not allow OXT to survive internalization [3], whereas the polar structure of Oxt may cause low membrane permeability [1]. A series of earlier investigations showed that IN OXT led to increased OXT concentrations at the level of CNS after nasal administration, in rhesus macaques [4], and in the amygdala and hippocampus in rats and mice [87]. However, a space to interpretations remained on whether actually synthetic or endogenous OXT constituted these increases [87], as nasal administration of OXT may indirectly generate the release endogenous OXT by stimulating hypothalamic OXT autoreceptors in a positive feedback loop [5]. Only recently, by using OXT knockout mice [1] showed in for the first time that IN administration of OXT permeates the extracellular fluid of specific areas of the brain, i.e. the amygdala and hippocampus, where it peaks between 30-60 minutes [1].
In our study we observed that IP OXT also induced improvements in behavioral state and reduced to a certain extent the oxidative stress. Interestingly, in the same study of [1], IP administration of OXT was also followed by increases in the OXT concentrations in the brain, though less significant with a shortened return to baseline when compared to the IN route [1].
We may presume this to be related to a potential action on the brain, either directly, which implies crossing of the BBB or indirectly. According to some studies, the central effects of IP OXT might be mediated through the OXTR, widely distributed in the periphery at the level of reproductive organs, heart or gastrointestinal tract, that activate the vagal afferent pathways and send signals to the brain [3,5]. Ferris et al., using special fMRI imaging and computational analysis to follow signals of central and peripheral OXT along integrated neural circuits in rats, found that IP administered OXT does not cross the BBB, yet triggers OXT neurotransmission in the cerebellum and several brainstem areas through sensory visceral afferents [90]. Also, a study in mice showed that the anxiolytic-like effects of peripherally administered OXT were blocked by a centrally administered OXTR antagonist which does not cross the BBB [88].
Regarding the issue of crossing the BBB, it is well-known that OXT is a relatively large neuropeptide (1008 Da) which seriously limits its passage ability [3]. However, only recently, Yamamoto et al. identified an uptake molecular mechanism for OXT to cross the BBB, represented by a member of the immunoglobulin receptors class, the receptor for advanced glycation end-products (RAGE) on brain capillary endothelial cells [91]. Direct OXT-RAGE binding was confirmed using multiple methods including surface plasmon resonance and gel permeation chromatography [91]. Cerebrospinal fluid (CSF) measures and microperfusion results showed that the OXT increases in the mice brain which follow exogenous (subcutaneous, intravenous or IN) administration were lost after RAGE knockdown. An interesting hypothesis of Correia Lima and Rodrigues states that OXT may enter the CNS at the level of several circumventricular organs, such as the vascular organ of lamina terminalis or neurohypophysis, that lack the tight junctions between the capillaries cells [6].
- Smith, A. S., Korgan, A. C., & Young, W. S. (2019). Oxytocin delivered nasally or intraperitoneally reaches the brain and plasma of normal and oxytocin knockout mice. Pharmacological Research, 104324.doi:10.1016/j.phrs.2019.104324
- Quintana, D. S., Guastella, A. J., Westlye, L. T., & Andreassen, O. A. (2015). The promise and pitfalls of intranasally administering psychopharmacological agents for the treatment of psychiatric disorders. Molecular Psychiatry, 21(1), 29–38. doi:10.1038/mp.2015.166
- Leng G, Ludwig M. Intranasal Oxytocin: Myths and Delusions. Biol Psychiatry. 2016 Feb 1;79(3):243-50.
doi: 10.1016/j.biopsych.2015.05.003 .
- Freeman, S.M.; Samineni, S.; Allen, P.C.; Stockinger, D.; Bales, K.L.; Hwa, G.G.; Roberts, J.A. Plasma and CSF oxytocin levels after intranasal and intravenous oxytocin in awake macaques. Psychoneuroendocrinolog. 2016, 66, 185–194
- D. Martins, N. Mazibuko, F. Zelaya, S. Vasilakopoulou, J. Loveridge, A. Oates, S. Maltezos, M. Mehta, M. Howard, G. McAlonan, D. Murphy, S. Williams, A. Fotopoulou, U. Schuschnig, Y. Paloyelis. Do direct nose-to-brain pathways underlie intranasal oxytocin-induced changes in regional cerebral blood flow in humans? bioRxiv 563056
- Correia Lima, J. P., & Rodrigues, A. L. (2019). Nasal Oxytocin: Facts and Routes. Acta Psychopathologica, 05(01). doi:10.4172/2469-6676.100181
